# Inhibition of *Salmonella* Binding to Porcine Intestinal Cells by a Wood-Derived Prebiotic

**DOI:** 10.3390/microorganisms8071051

**Published:** 2020-07-15

**Authors:** Aleksandar Božić, Robin C. Anderson, Tawni L. Crippen, Christina L. Swaggerty, Michael E. Hume, Ross C. Beier, Haiqi He, Kenneth J. Genovese, Toni L. Poole, Roger B. Harvey, David J. Nisbet

**Affiliations:** 1Faculty of Agriculture, University of Novi Sad, Novi Sad 21000, Serbia; aleksandar.bozic@stocarstvo.edu.rs; 2Food and Feed Safety Research Unit, United States Department of Agriculture/Agricultural Research Service, College Station, TX 77845, USA; tc.crippen@usda.gov (T.L.C.); christi.swaggerty@usda.gov (C.L.S.); Michael.Hume@usda.gov (M.E.H.); Ross.Beier@usda.gov (R.C.B.); Haiqi.He@usda.gov (H.H.); Kenneth.Genovese@usda.gov (K.J.G.); Toni.Poole@usda.gov (T.L.P.); Roger.Harvey@usda.gov (R.B.H.); David.Nisbet@usda.gov (D.J.N.)

**Keywords:** oligosaccharide, *Salmonella Typhimurium*, wood-derived prebiotic

## Abstract

Numerous *Salmonella*
*enterica* serovars can cause disease and contamination of animal-produced foods. Oligosaccharide-rich products capable of blocking pathogen adherence to intestinal mucosa are attractive alternatives to antibiotics as these have potential to prevent enteric infections. Presently, a wood-derived prebiotic composed mainly of glucose-galactose-mannose-xylose oligomers was found to inhibit mannose-sensitive binding of select *Salmonella*
*Typhimurium* and *Escherichia coli* strains when reacted with *Saccharomyces boulardii*. Tests for the ability of the prebiotic to prevent binding of a green fluorescent protein (GFP)-labeled *S.*
*Typhimurium* to intestinal porcine epithelial cells (IPEC-J2) cultured in vitro revealed that prebiotic-exposed GFP-labeled *S.*
*Typhimurium* bound > 30% fewer individual IPEC-J2 cells than did GFP-labeled *S.*
*Typhimurium* having no prebiotic exposure. Quantitatively, 90% fewer prebiotic-exposed GFP-labeled *S.*
*Typhimurium* cells were bound per individual IPEC-J2 cell compared to non-prebiotic exposed GFP-labeled *S.*
*Typhimurium*. Comparison of invasiveness of *S.*
*Typhimurium* DT104 against IPEC-J2 cells revealed greater than a 90% decrease in intracellular recovery of prebiotic-exposed *S.*
*Typhimurium* DT104 compared to non-exposed controls (averaging 4.4 ± 0.2 log_10_ CFU/well). These results suggest compounds within the wood-derived prebiotic bound to *E. coli* and *S.*
*Typhimurium*-produced adhesions and in the case of *S.*
*Typhimurium*, this adhesion-binding activity inhibited the binding and invasion of IPEC-J2 cells.

## 1. Introduction

*Salmonella enterica* comprises many zoonotic bacterial serovars that can cause serious life-threatening enteric and systemic diseases in infected hosts [1]. *Salmonella* can also establish an asymptomatic carrier state in infected animals and in food animals such as swine which can make detection and diagnosis of infected animals difficult thereby leading to missed opportunities for treatment during rearing or for removal of infected animals from the slaughter queue prior to processing. Consequently, a need exists for treatments or strategies to prevent *Salmonella* infections in food animals. Whilst antibiotics have traditionally been and continue to be used to treat and prevent animal diseases, their use as food safety interventions to prevent bacterial contamination of carcasses is not an accepted application. Moreover, public health agencies are concerned that the emergence and dissemination of antimicrobial resistant bacteria in animal agriculture may render antibiotics less effective in humans. This has increased efforts to develop non-antibiotic therapies for prevention and control of infectious diseases. One attractive non-antibiotic strategy for preventing enteric *Salmonella* infections in food animals is to feed complex carbohydrate-rich prebiotics to promote the proliferation of healthy gut microflora able to outcompete or inhibit *Salmonella* colonization in the gut. Mechanistically, as reviewed recently by Liu et al. [2], the utilization of prebiotic carbohydrates as fermentative substrates has been reported to promote beneficial lower gut fermentation resulting in increased fermentation acid production, with some acids such as butyrate being stimulatory for villus growth and proliferation. Other beneficial effects are also reported for prebiotics in swine health resulting from the selection of certain beneficial bacterial populations and enhancement of the host’s immune status [2,3]. Alternatively, in some cases prebiotics containing complex carbohydrate-rich products such as mannan- or galactomannan- or other carbohydrate-substituted oligosaccharides have been reported to block adherence of pathogens such as *Salmonella* and enteropathogenic *Escherichia coli* to the animal’s gut mucosa thereby preventing the initiation of colonization and infection within the animal [4].

Conceptually, these anti-binding approaches attempt to use soluble monosaccharides or oligosaccharides within a prebiotic product to bind to bacterial surface-associated lectins thereby interfering with the bacteria’s ability to attach to mucosal-associated attachment sites, the latter often composed of glycoproteins post-translationally modified with structurally similar carbohydrate-based substituent groups [5]. Lectin-attachment interactions are specific to certain protein appendages such as fimbriae and oftentimes their binding interactions with oligosaccharides may be specific to the type of sugar molecule making up the oligosaccharide. Recently, a wood-derived product extracted with steam from Southern Yellow Pine chips and composed mainly of glucose-galactose-mannose-xylose oligomers has been investigated as a prebiotic for shrimp, dogs and broilers [6,7,8,9]. Beneficial prebiotic properties such as promoting increased gut populations of *Bifidobacterium* and *Lactobacillus* spp. and potentially enhancing gut immune function were observed in these studies, however, the expected benefits of inhibiting intestinal colonization by *Salmonella* were not observed. Moreover, the adherence-blocking ability of the wood-derived prebiotics has not yet been reported. Accordingly, the main objective of the present study was to test if this potential prebiotic may bind bacterial surface-associated lectins thereby interfering with the binding of *Salmonella enterica* serovar Typhimurium and *E. coli* to yeast or intestinal cells.

## 2. Material and Methods

### 2.1. Bacteria

*Salmonella enterica* serovars Typhimurium DT104 (DHEP 12362), Typhimurium NVSL-1776 of swine origin and an undesignated poultry source Typhimurium were stock cultures used previously [10,11,12]. The green fluorescent protein labeled (GFP)-labeled *Salmonella Typhimurium* was graciously provided by Drs. Roy J. Bongaerts and Jay Hinton (Norwich Research Park, Norwich, UK) [13]. *Escherichia coli* strains designated CVM 1585, 1582, 1569, 832 were stock cultures used earlier [14] expressing F4, F5, F6 and F107, respectively, and had graciously been provided to us by Dr. Nancy Cornick at Iowa State University (Ames, IA, USA). *Escherichia coli* strain RCA-1 was a non-fimbriated bovine source isolated by us and confirmed to contain no amplifiable fimbrial sequences when characterized by the *E. coli* Reference Center (Penn State College of Agricultural Sciences, University Park, PA, USA). The bacteria were resuscitated from stocks preserved in cryo-beads containing 20% glycerol at −80 °C by plating to Brilliant Green Agar (Oxoid, Unipath Ltd., Basinstoke, Hampshire, UK) plates supplemented with 20 µg novobiocin/mL (BGA-N) or 12 µg chloramphenicol/mL (BGA-C) for the respective *Salmonella* strains. The plates were incubated at 37 °C for 24 h and well isolated colonies were re-streaked the day before use on BGA-N or BGA-C plates or inoculated into Tryptic Soy broth (Becton Dickinson, Sparks, MD, USA) and incubated overnight at 37 °C. For use in slide agglutination studies, bacteria were harvested from overnight grown cultures (10 mL) via centrifugation (15 min at 10,000× *g*), washed once (10 mL) and then resuspended with phosphate buffered saline (PBS, pH 7.4) to achieve 10 ^8^ colony forming units (CFU)/mL as determined by viable cell count. For use in adherence or invasiveness studies, colonies developed overnight were transferred to PBS and concentrations of bacteria in the stock bacterial suspensions were monitored by spectrophotometry until achieving an optical density (625 nm) of 0.7 which a upon viable cell count was approximately 10 ^8^ CFU/mL.

### 2.2. Prebiotic

The wood-derived prebiotic product was provided as a dried product by Temple Inland (Diboll, TX, USA) prior to being acquired by International Paper (Memphis, TN, USA) in 2012. The product contained approximately 90% carbohydrate, mainly in oligosaccharide form with free monosaccharaides being less than 9% of the product (mainly arabinose, xylose and galactose), with lesser amounts of fucose, rhamnose and fructose [7]. When hydrolyzed for compositional analysis, the predominant sugars were mannose, glucose, xylose, galactose, arabinose, rhamnose and fucos present at 35, 16, 13, 8, 4, 0.6 and 0.3%, respectively [9].

### 2.3. Effects on Binding Activity via Slide Agglutination Tests

To test the effects of the experimental wood-derived prebiotic on binding activity of *S. Typhimurium* DT104 and the GFP-labeled *S. Typhimurium* and select strains of *Escherichia coli*, a qualitative slide agglutination test was conducted as described by Mirelman et al. [15] except using *Saccharomyces boulardii* (Biocodex, San Bruno, CA, USA) instead of *Saccharomyces cerevisiae* (Fleishman’s, Memphis, TN, USA) as the host [16]. This substitution in yeast species was done because commercially acquired *Saccharomyces cerevisiae* cells did not support agglutination. The yeast cells were grown in medium containing (*wt/v*) 1% yeast extract and 2% each of peptone (Becton Dickinson, Sparks, MD, USA) and glucose (Sigma-Aldrich, St. Louis, MO, USA) and were pelleted via centrifugation (15 min at 10,000× *g*). The harvested yeast cells were washed once with PBS, pelleted again via centrifugation (15 min at 10,000× *g*) and resuspended in fresh buffer to a final concentration of approximately 0.1 *g* wet wt/mL.

Agglutination tests were performed on an microscope slide by combining 50 µL of bacterial suspension that had just previously been pretreated (1:1) with water, the wood-derived prebiotic or 0.05 M methyl α-d-mannopyranoside (the latter included as a positive control) with 50 µL of the yeast suspension. Concentrations of bacterial cells and yeast were added to achieve ratios that just barely caused an observable agglutination reaction when mixed without added competitive-binding substrate. These optimal ratios were determined in preliminary studies via mixing of 50 µL volumes of serial 10-fold dilutions of the bacterial and yeast cells suspensions. Upon exposure to the yeast cells, bacterial cells exhibiting profuse cell clumping within 3 min exposure were interpreted as a positive agglutination reaction. Based on an estimated 35% mannose content, mixing 50 µL of the wood-derived prebiotic prepared at 10, 20, 40 or 80 mg/mL would have exposed the bacterial cells to the equivalent of 0.02, 0.04, 0.08 or 0.16 M mannose. Reactions showing agglutination were interpreted as bacterial cells binding to yeast cells whereas results showing non-agglutination were interpreted as the inability of the bacterial cells to bind to the yeast cells, due either to prebiotic-caused inhibition or the absence of functional fimbriae.

### 2.4. Effects on Adherence of a GFP-Labeled S. Typhimurium to Intestinal Porcine Epithelial Cells

A non-immortalized porcine jejunal cell line (IPEC-J2) was obtained from Dr. Radhey S. Kaushik (South Dakota State University, Brookings, SD, USA). These jejunal cells have been used in studies to examine non-mannose sensitive adherence characteristics of different enterotoxigenic *Escherichia coli* strains affecting pigs and have been shown to support adherence by *E. coli* expressing K88 (also called F4), K99 (also called F5) or F41 [17] fimbriae. These cells also supported the adherence and internalization of *S. Typhimurium* DT104 [18,19,20].

For these studies, IPEC-J2 cells harvested from cultures grown at 37 °C with 5% CO_2_ in Dulbecco’s Modified Eagle Medium (DMEM + Hams F-12; Invitrogen Corporation, Grand Island, NY, USA) supplemented with 5% fetal calf serum (Atlanta Biologicals, Lawrenceville, GA, USA), 100 IU penicillin/mL, 100 µg streptomycin/mL (Invitrogen), 5 µg insulin/mL and 5 ng epidermal growth factor/mL (Sigma-Aldrich) were seeded to 12-well chamber slides (Nalge Nunc International, Roskilde, Denmark) by addition of 250 to 300 µL of a suspension containing 4.3 ± 2.6 × 10 ^5^ IPEC-J2 cells. The chamber slides were incubated as above and replenished with fresh medium at 2-day intervals for approximately 1 week and replaced with medium lacking antibiotics and calf-serum the day before challenge with the GFP-labeled *S.* Typhimurium. During each day of testing, residual medium was removed from all wells of each chamber slide. Suspensions containing the GFP-labeled *S. Typhimurium* (250 µL DEM + Hams F-12 medium lacking antibiotics and calf and previously exposed for 10 min to an equal volume of medium alone or to suspensions containing 18 mg wood-derived prebiotic product/mL (the equivalent of 0.1 M d-mannose) or 0.05 M methyl α-d-mannopyranoside were added to the different wells of the 8-well chamber slides. In initial adherence studies, the bacterial suspensions were added to chamber slides to provide a challenge of approximately 10 ^5^, 10 ^6^, 10 ^7^ or 10 ^8^ GFP-labeled *S. Typhimurium* cells/mL. In later studies, the cells were challenged with 10 ^5^ to 10 ^7^ bacteria/mL and this resulted in sufficient binding for enumeration. After addition of treated or untreated bacteria, the chamber slides were incubated 3 h at 37 °C to allow attachment to occur and the wells of each slide were washed 3× with 400 µL PBS and examined by phase contrast microscopy at 1000× magnification. Controls and treatments at each challenge level were each replicated together, 2 wells per treatment, on three separate chamber slides and numbers of attached GFP-labeled *S.* Typhimurium were counted on 100 randomly selected IPEC-J2 cells per each well.

### 2.5. Effect on Invasiveness of S. Typhimurium DT104 for Intestinal Porcine Epithelial Cells

Using an invasive protocol similar to that conducted by He et al. [21], the IPEC-J2 cells grown as described above were seeded in Falcon 12-multiwell plates (Becton Dickinson Labware, Franklin Lakes, NJ, USA) and grown overnight in DMEM + Hams F-12 lacking antibiotics and calf-serum. The morning of the test, cells were inoculated with approximately 10 ^5^ CFU *S.* Typhimurium DT104/mL having been pre-exposed for 10 min to 18 or 36 mg wood-derived prebiotic/mL (equivalent of 0.1 or 0.2 M d-mannose) or to 0.1 or 0.2 M methyl α-d-mannomanpyranoside. The plates were incubated for 1 h at 37 °C under CO_2_ and the cells were gently washed by addition and removal of fresh DMEM + Hams F-12 lacking antibiotics or serum, treated with 100 µg gentamycin sulfate/mL (in DMEM + Hams F-12) and incubated for an additional hour to kill non-internalized *S. Typhimurium* DT104. The IPEC-J2 cells were washed 2× with PBS after gentamycin treatment then subjected to bacteriological enumeration via plating of serial 10-fold dilutions of cell lysed by 1% Triton X-100 in PBS to Brilliant Green agar supplemented with 20 µg novobiocin/mL.

### 2.6. Statistical Analysis

Quantitative measures of bacterial adherence (numbers of GFP-labeled *S. Typhimurium* bound/IPEC cell) and qualitative measures of adherence (proportion of IPEC cells bound by at least one GFP-labeled *S. Typhimurium*) were tested at each bacterial challenge level for effects of the wood-derived prebiotic by a completely randomized analysis of variance and a Two-sided Dunnett’s multiple comparison to controls using Statistix 9 Analytical Software (Statistix, Tallahassee, FL, USA). Quantitative measures of bacterial intracellular survival (numbers of *S. Typhimurium* DT104 recovered from lysed IPEC-J2 cells) were analyzed likewise.

## 3. Results and Discussion

### 3.1. Effects on Binding Activity via Slide Agglutination Tests

All strains of *S. Typhimurium* exhibited agglutination to *Saccharomyces boulardii* and this activity was overcome when cells were treated with 0.05 M methyl α-d-mannopyranoside or between 20 to 80 mg wood-derived prebiotic/mL (Table 1). Results from the qualitative slide agglutination tests in this study also revealed that *E. coli* CVM strains 1569 and 1585 expressing F4 (also referred to as a K88 type fimbria) or F6 (the latter also referred as 987P) fimbria, respectfully, exhibited agglutination activity to *Saccharomyces boulardii* and this agglutination activity was likewise overcome when cells were treated with 0.05 M methyl α-d-mannopyranoside or as little as 20 mg wood-derived prebiotic/mL (Table 1). The type of adherence to yeast cells observed with mannose-sensitive bacteria is thought to be mediated by Type 1 fimbriae expressed by the bacteria. While it is attractive to speculate that inhibition of agglutination observed in mixtures treated with the wood-derived prebiotic may reflect a protective effect of free or conjugated mannose constituents with the prebiotic it cannot be ruled out that other fimbrial adhesions may be involved in the agglutination. For instance, Koh et al. [17] reported competent binding of *E. coli* strains expressing K88-type adhesions as well as an *E. coli* strain expressing both K99 and F41, but not a strain expressing only K99, to IPEC-J2 cells when reacted even in the presence of methyl α-d-mannopyranoside added to prevent Type 1 mediated binding. Mechanistically, F4 fimbria are known to bind to galactosyl-substituted glycolipids and to F6 fimbria, which may be structurally similar to Type 1 fimbriae. F4 fimbria also bind to fucosyl- and galactosyl-substituted glycoproteins [22]. Accordingly, further study is needed to ascertain if galactose or fucos moieties present within the wood-derived prebiotic may be contributing to the inhibition of agglutination. *Escherichia coli* strains containing an F5 or an F107 (also called F18) fimbriae (strains 1582 and 832, respectively) and *E. coli* RCA-1, which is a nonfimbriae-expressing strain did not or only very weakly exhibited mannose-sensitive adhesion in this study. *Escherichia coli* expressing F5 or F107 fimbria appear to exhibit specificity to receptors presented by porcine epithelial cells and thus it is not unexpected these strains, as well as the nonfimbriated strains failed to agglutinate the yeast cells.

### 3.2. Effects on Adherence of a GFP-Labeled S. Typhimurium to Intestinal Porcine Epithelial Cells

Tests with the GFP-labeled *S. Typhimurium* revealed this bacterium readily bound the IPEC-J2 cells and this binding was qualitatively inhibited when the bacterium had been pre-exposed for 10 min to the wood-derived prebiotic (Figure 1). Quantitatively, we found that pre-exposure of the GFP-labeled *S.* Typhimurium to 18 mg of the wood-derived prebiotic/mL, which based on results from Table 1 may be near or at saturating levels, decreased (*p* < 0.05) numbers of GFP-labeled *S. Typhimurium* bound to the IPEC-J2 cells as well as the proportion of IPEC-J2 cells bound by the GFP-labeled *S. Typhimurium* (Figure 2). Conversely, similar pre-exposure of the GFP-labeled *S. Typhimurium* to 0.05 M methyl α-d-mannopyranoside for 10 min had no effect (*p* > 0.05) on GFP-labeled *S. Typhimurium* binding to the IPEC-J2 cells when inoculated with 10 ^6^ or 10 ^7^ CFU bacteria (not shown), with 4.02 ± 2.5 and 5.8 ± 1.6 vs. 2.8 ± 1.2 and 10.4 ± 3.7 CFU mannose-exposed and nonexposed GFP-labeled *S. Typhimurium* observed bound per each IPEC-J2 cell counted, respectively. This observation indicates that adherence of the GFP-labeled *S. Typhimurium* to the IPEC cells may have occurred via mechanisms independent of mannose-specific adhesions. It has been reported that Type 1 fimbrial mediated mannose-sensitive binding by *Salmonella* does not represent the only type of bacterial-host cell interactions that can occur and has been reported to occur in only about 48% of measured strains [15]. There are, however, other fimbrial adhesions expressed by *Salmonella* can recognize and bind to other receptors substituted with fucosyl-, ganglioside GM1, sialic acid, *N*-acetylgalactosamine and galactosyl-moieties presented on the outside of mammalian host cells [23,24] and in some cases this binding can be inhibited by free lactose, galactose and xylose as well as galacto- and xylooligosaccharides [25,26]. Moreover, it has been reported that branched-chain tri- or greater branched chain mannan-oligosaccharides configured with α-glycosidic linkages exhibited greater specificity and sensitivity against tested *E. coli* fimbriae, although such specificity was less stringent against *S. Typhimurium* [27]).

### 3.3. Effect on Invasiveness of S. Typhimurium DT104 to Intestinal Porcine Epithelial Cells

Whereas *Salmonella* can encode and express their own proteins to allow the invasion of host cells, evidence suggests they use fimbriae or other adhesions to recognize specific carbohydrate determinants to initiate binding on susceptible host cells [28]. Therefore, it is possible that the anti-adhesion compounds present within the wood-derived prebiotic may also interfere with the invasive ability of *Salmonella*. Accordingly, this was tested using an invasive *S. Typhimurium* DT104 strain in a cell invasion assay with the IPEC-J2 cells.

Consistent with earlier reports, [19,20], *S. Typhimurium* DT104 cells having no exposure to the wood-derived prebiotic or methyl α-d-mannopyranoside readily invaded the IPEC-J2 cells (Figure 3). However, invasion was decreased (*p* < 0.05) when the *S. Typhimurium* DT104 cells were exposed to 18 or 36 mg/mL (equivalent to 0.1 or 0.2 M mannose equivalents) of the wood-derived prebiotic as reflected by decreased recovery of intracellular *S. Typhimurium* DT104 (Figure 3). As expected, invasion of the IPEC-J2 cells by *S. Typhimurium* DT104 cells pre-exposed to 0.1 or 0.2 M methyl α-d-mannopyranoside was similarly decreased (*p* < 0.05), which suggests in this case that mannose-sensitive adhesions contributed to the inhibition of invasion.

## 4. Conclusions

A wood-derived prebiotic product obtained by steam extraction from Southern Yellow Pine chips and composed mainly of glucose-galactose-mannose-xylose oligomers was shown to inhibit binding of certain enteropathogenic *E. coli* and *S. Typhimurium* strains to *Saccharomyces boulardii* in a slide agglutination assay and the binding and invasion of select *S. Typhimurium* to IPEC-J2 cells but whether or not the anti-binding activity contributed to colonization resistance within the host is not readily apparent. Whereas performance enhancing effects of various oligosaccharide products have been observed in swine and poultry, these have most often been attributed to immuno-stimulation and enrichment of beneficial bacterial populations with few reports of inhibition of *Salmonella* colonization [9,29]. From a practical perspective, a major limitation of the anti-lectin activity of the oligosaccharides in animals is that the sugars are likely degraded by diverse populations of carbohydrate-degrading bacteria inhabiting the gastrointestinal tract. For instance, Farber et al. [7,30] had shown that the carbohydrates in the wood-derived prebiotic were readily fermented by gut bacteria from a mature canine but did not differentiate between the different carbohydrate-linked oligosaccharides. Naughton et al. [31] observed significant inhibition of *Salmonella* and *E. coli* in porcine ileal and jejunal tissues, respectively, when freshly harvested and then exposed to galacto-oligosaccharide treatment and bacterial challenges in vitro but not when the prebiotics were fed to the animals. The diminished protective effect of the galacto-oligosaccharide treatments was attributed to intestinal fermentation. Conversely, the protective effect of fructo-oligosaccharide treatments on *Salmonella* binding were observed with both their in vitro model as well as with their pigs fed the fructo-oligosaccharide and this was because the fructo-oligosaccharide was less degradable than the galacto-oligosaccharide by the gut bacteria [31]. Tran et al. [32] reported that xylooligosaccharides were fermented, albeit slowly, when tested in an in vitro fermentation model using mixed populations of porcine gut bacteria. Accordingly, further studies are needed to determine how best to realize the anti-binding activity of the wood-derived prebiotic which may be most impactful if administered to suckling piglets not yet possessing established populations of gut bacteria or to support for antibiotic or other chemotherapies administered to pigs with established gut flora. Conceptually, certain combined therapies may be able to limit the activity of saccharolytic (prebiotic-degrading) bacterial populations thereby preserving the adhesion-binding activity of the oligosaccharides. Additionally, it is reasonable to hypothesize that short-term administration of natural glucosidase inhibitors [33,34,35] during periods when pigs may be most susceptible to *Salmonella* or *E. coli* infection may temporarily prevent degradation and subsequent loss of the lectin-binding activity by saccharolytic bacterial populations. The inhibition of *S*. Typhimurium invasion potentially achieved when the wood-derived prebiotic is combined with appropriate antibiotic therapy may also aid the chemotherapeutic activity by preventing pathogen sequestration within intestinal cells, the latter being postulated to be a mechanism by which pathogens can evade exposure to the antibiotic [36].

## Figures and Tables

**Figure 1 microorganisms-08-01051-f001:**
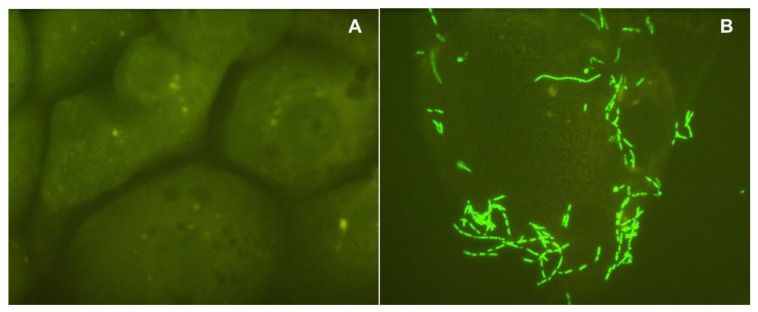
Phase contrast image (1000× magnification) of non-immortalized intestinal porcine derived jejunal cells (IPEC-J2) challenged with 10 ^8^ cells of a green fluorescence protein (GFP)-labeled *Salmonella Typhimurium* previously exposed for 10 min to 0 or 18 mg wood-derived prebiotic/mL of medium. Results show that IPEC-J2 cells challenged with the GFP-labeled *S. Typhimurium* cells previously exposed to the wood-derived prebiotic product were qualitatively bound by far fewer GFP-labeled *S. Typhimurium* (**A**) than IPEC-J2 cells challenged with GFP-labeled *S. Typhimurium* having no exposure to the wood-derived prebiotic product (**B**).

**Figure 2 microorganisms-08-01051-f002:**
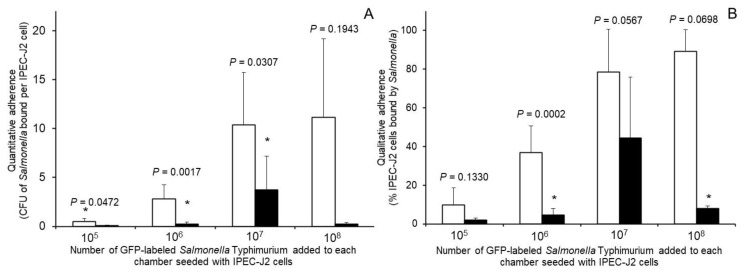
Effect of pre-exposing green fluorescent protein (GFP)-labeled *Salmonella Typhimurium* (black bars) with 18 mg wood-derived prebiotic/mL of medium on this bacterium’s quantitative (**A**) and qualitative (**B**) adherence to a non-immortalized porcine derived jejunal cell line (IPEC-J2). Control GFP-labeled *S. Typhimurium* exposed to medium alone are presented with white bars. Asterisks denote decreased adhesion (*p* < 0.05) of the GFP-labeled *S**. Typhimurium* that had been pre-exposed to the wood-derived prebiotic when compared to its control.

**Figure 3 microorganisms-08-01051-f003:**
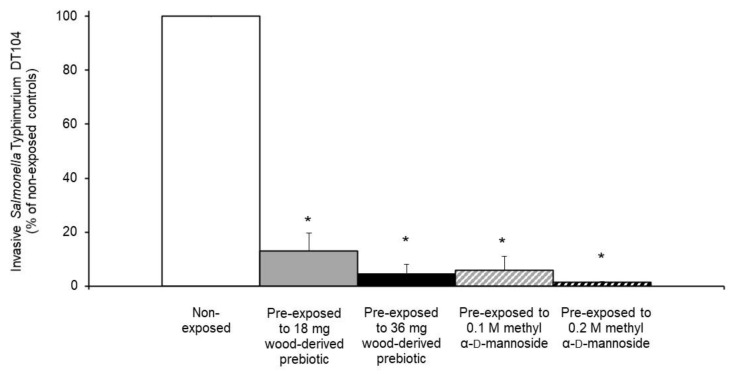
Effect of pre-exposing *Salmonella* Typhimurium DT104 cells to 18 or 36 mg wood-derived prebiotic/mL (solid gray or black bars, respectively) or to 0.1 or 0.2 M methyl α-D-mannopyranoside (slashed gray or black bars, respectively) per mL exposure medium on quantitative measures of invasion within non-immortalized porcine derived jejunal cell line (IPEC-J2). Control *S. Typhimurium* DT104 cells pre-exposed to medium alone were recovered at 4.4 × 0.2 log_10_ CFU/well and are presented as 100% with white bars. Asterisks denote a decrease in invasion (*p* < 0.05) of the porcine intestinal cells by pre-exposed *S. Typhimurium* DT104 when compared to controls.

**Table 1 microorganisms-08-01051-t001:** Effect of wood-derived prebiotic on fimbrial-binding of select *Salmonella enterica* and *Escherichia coli* strains to *Sacchaomyces boulardii.*

-	Effect of Wood-Derived Prebiotic (mg/mL) on Agglutination Reaction ^a^			
Bacteria	10	20	40	80
*Salmonella* Typhimurium (poultry isolate)	+++	+++	++	-
*Salmonella* Typhimurium NVSL 95-1776 (swine isolate)	+++	++	-	-
*Salmonella* Typhimurium DT104	+++	++	-	-
*Salmonella* Typhimurium (GFP-labeled)	+++	-	-	-
*Salmonella* Typhimurium (unlabeled parent)	+++	-	-	-
*Escherichia coli* CVM 1569 (F6)	+++	-	-	-
*Escherichia coli* CVM 1585 (F4)	+++	-	-	-

^a^ Agglutination reflects binding of bacteria to *Saccharomyces boulardii* cells. Symbols +++ denote strong agglutination reaction; ++ denote a mild agglutination reaction and – denotes no observable agglutination. Mixtures exhibiting agglutination are interpreted as causing no or mild inhibition of binding between bacteria and yeast cells whereas no agglutination reactions are interpreted as appreciable inhibition of binding between bacteria and yeast cells. All bacteria exhibited inhibition of mannose-sensitive binding to *Saccharomyces boulardii* cells after pre-exposed similarly to 0.05 M methyl α-d-mannopyranoside.

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
