# Peer review of "Inhibition of Salmonella Binding to Porcine Intestinal Cells by a Wood-Derived Prebiotic"

_microorganisms, 2020, doi:10.3390/microorganisms8071051_

Round 1

Reviewer 1 Report

In this paper authors discuss effects of wood-derived prebiotics composed mainly of glucose-galactose-mannose-xylose oligomers on invasiveness and binding of S. Typhimurium to intestinal porcine epithelial cells. Although findings are interesting, my concerns are:

  1. In lines 182 and 183 / 187 and 188 you claim that activity of agglutination was overcome when cells were treated with as little as 10 mg wood-derived probiotic/mL (Table 1) but in the table 1 I can see that at that concentration agglutination is still present for all used strains of S. Typhimurium and E. coli.
  2. How do you explain that pre-exposing green fluorescent protein (GFP)-labeled Salmonella Typhimurium 247 with 18 mg wood-derived prebiotic/mL of medium had higher influence on adherence to a IPEC-J2 when you used higher number of S. Typhimurium (Figure 2; 107 and 108)

Author Response

Authors' response: 

Reviewer 2 Report

This paper reports upon results of in vitro experimentation of testing Salmonella and E. coli binding to yeast cells in the presence and absence of a wood-derived polysaccharide. The authors propose that the polysaccharide could be used as a prebiotic to “promote the proliferation of healthy gut microflora,” “potentially enhancing gut immune function,” and prevent enteric Salmonella infection. However, the results/findings in the paper do not answer this question or support this statement or describe how the prebiotic would be administered to reach the appropriate gut or intestinal target. The authors discuss the work by Farber et al. that showed “the carbohydrates in the wood-derived prebiotic were readily fermented by gut bacteria” which underscores that moving from in vitro to in vivo will not be trivial. The results of the experimentation are interesting and are worthy of publication but the introduction and focus of the paper should be modified to reflect the goals and the experiments in the study.

Was the wood-derived prebiotic structurally characterized? The authors state it is “composed mainly of glucose-galactose-mannose-xylose oligomers.” A figure showing the structure of the carbohydrate polymer would be helpful as well as a comparison to the structure of the mannopyranoside control.

Overall, the paper is well-written but I did find a few minor edits:

Author Roger B. Harvey is missing an institutional affiliation.

Use the degree symbol for 37 degrees and not a superscript o.

Include City and State of supplies of reagents (e.g., Becton Dickinson).

Page 4, Line 160: Decton Dickinson should be “Becton Dickinson”.

Disclosure statement: “no conflicting” instead of “not conflicting”
